

# Selective concentration of iron, titanium, and zirconium substrate minerals within Gregory's diverticulum, an organ unique to derived sand dollars (Echinoidea: Scutelliformes)

Louis G. Zachos[1] and Alexander Ziegler[2]

[1] Department of Geology and Geological Engineering, University of Mississippi, Oxford, Mississippi, United States

[2] Bonner Institut für Organismische Biologie, Rheinische Friedrich-Wilhelms-Universität, Bonn, Germany

Corresponding author
Louis G. Zachos,
lgzachos@olemiss.edu

## ABSTRACT

Gregory's diverticulum, a digestive tract structure unique to a derived group of sand dollars (Echinoidea: Scutelliformes), is filled with sand grains obtained from the substrate the animals inhabit. The simple methods of shining a bright light through a specimen or testing response to a magnet can reveal the presence of a mineral-filled diverticulum. Heavy minerals with a specific gravity of >2.9 g/cm$^3$ are selectively concentrated inside the organ, usually at concentrations one order of magnitude, or more, greater than found in the substrate. Analyses of diverticulum content for thirteen species from nine genera, using optical mineralogy, powder X-ray diffraction, scanning electron microscopy and energy dispersive X-ray spectroscopy, as well as micro-computed tomography shows the preference for selection of five major heavy minerals: magnetite ($Fe_3O_4$), hematite ($Fe_2O_3$), ilmenite ($FeTiO_3$), rutile ($TiO_2$), and zircon ($ZrSiO_4$). Minor amounts of heavy or marginally heavy amphibole, pyroxene and garnet mineral grains may also be incorporated. In general, the animals exhibit a preference for mineral grains with a specific gravity of >4.0 g/cm$^3$, although the choice is opportunistic and the actual mix of mineral species depends on the mineral composition of the substrate. The animals also select for grain size, with mineral grains generally in the range of 50 to 150 μm, and do not appear to alter this preference during ontogeny. A comparison of analytical methods demonstrates that X-ray attenuation measured using micro-computed tomography is a reliable non-destructive method for heavy mineral quantification when supported by associated analyses of mineral grains extracted destructively from specimens or from substrate collected together with the specimens. Commonalities in the electro-chemical surface properties of the ingested minerals suggest that such characteristics play an important role in the selection process.

## INTRODUCTION

The distinctive group of echinoids known familiarly as sand dollars is globally distributed along tropical to temperate coasts, and the flattened morphology is adapted to sandy, shallow littoral waters exposed to wave and tidal current energy (*Smith, 1984*). The sand dollars are divided into the Laganiformes and Scutelliformes, considered to be either sister (*Mongiardino Koch & Thompson, 2021*; *Mongiardino Koch et al., 2022*) or polyphyletic (*Lee et al., 2023*) clades. Scutelliformes are often considered to be "true" sand dollars (*Mongiardino Koch et al., 2018*). Most extant scutelliform sand dollar species possess a soft tissue organ derived from the digestive tract known as Gregory's diverticulum (*Gregory, 1905*; *Mitchell, 1972*; *Lawrence, 2001*; *Ziegler & Barr, 2018*; *Ziegler, 2023*). This structure is a transparent, blindly ending, ramified tube attached to the rectum *via* a short canal, the duct connector (Fig. 1). Prior to sexual maturity, the animals fill the diverticulum with mineral grains picked up from the substrate (*Mitchell, 1972*; *Chia, 1973*). During ontogeny, the mineral grains are usually entirely expelled from a given point on and the diverticulum then atrophies (*Ziegler et al., 2016*). The earliest interpretation of this behavior suggested that the mineral grains have a ballast function, literally acting as a weight belt intended to render the animals heavier and improve hydrodynamic stability (*Chia, 1973*). A mineral-grain-filled diverticulum occurs in (and is restricted to) a well-defined group of scutelliforms (*Linder, Durham & Orr, 1988*; *Mooi & Chen, 1996*; *Ziegler et al., 2016*). A number of studies have found that there is a strong preference for the presence of heavy minerals with a specific gravity greater than 2.9 g/cm$^3$ inside the diverticulum. For example, *Dendraster excentricus* and *Mellita quinquiesperforata* both select for iron oxide minerals (*Chia, 1973*, *1985*; *Borzone, Tavares & Soares, 1997*), while *Scaphechinus mirabilis* accumulates zircon and ilmenite at much higher concentrations than found in the substrate (*Elkin et al., 2012*; *Begun et al., 2014*). However, one study reported that *Sinaechinocyamus mai* selected grains with approximately the same proportion of light to heavy minerals as found in the substrate (*Chen & Chen, 1994*). In addition, other studies have suggested that minerals are degraded in the diverticulum or the intestine instead of being expelled (*Chia, 1985*; *Elkin et al., 2013*). Unfortunately, nearly all previous mineralogical studies of mineral grains found inside the organ as well as in the substrate were based on optical methods using a petrographic microscope. In only a single case (*Elkin et al., 2012*) the optical identifications were verified using energy dispersive X-ray spectroscopy (EDS).

In order to provide a comprehensive picture of mineral selection in scutelliform sand dollars, the primary objective of the present study was to delineate any patterns in the selection of mineral grains that might occur. In particular, we set out to understand how the type and proportion of minerals in Gregory's diverticulum are related to those in the substrate. In addition, a further objective of the present study was to evaluate the precision and accuracy of different analytical methods, both destructive and non-destructive, to identify the ingested minerals.

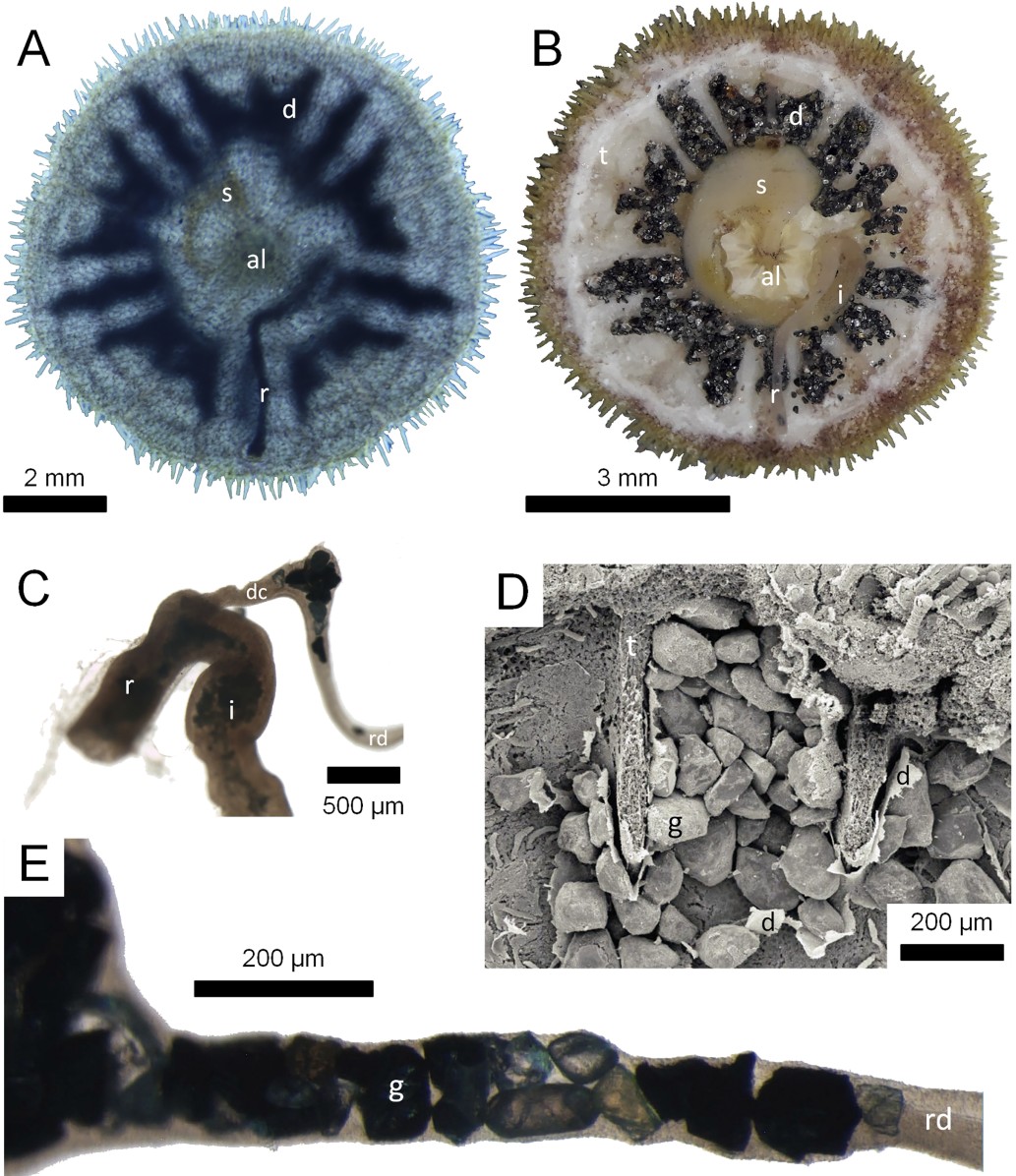

**Figure 1  Morphological aspects of Gregory's diverticulum in *Echinarachnius parma*.** (A) Aboral view of juvenile specimen with transmitted light exposing selected internal structures such as Aristotle's lantern (al), stomach (s), rectum (r), and Gregory's diverticulum (d). (B) Aboral view of juvenile specimen with aboral half of test (t) dissected away to show internal organs, including intestine (i). (C) Close-up of connection between intestine and rectum with duct connector (dc) branching off, followed by the ring duct (rd). Note sediment inside the intestine and mineral grains inside the ring duct. (D) SEM micrograph of mineral grains (g) inside Gregory's diverticulum partially dissected open. (E) Close-up of mineral grains inside the ring duct.     

## MATERIALS AND METHODS

### Specimens

Various sources supplied juvenile specimens of extant sand dollars pertaining to thirteen species from nine scutelliform genera known to possess Gregory's diverticulum (Table 1). Field collection in Costa Rica was approved by the National System of Conservation Areas (SINAC-SE-CUSBSE-PI-R-131-2016) as well as in Mexico by the Comisión Nacional de Acuacultura y Pesca (PPF/DGOPA-291/17). Species identifications were made based on the most current taxonomic treatments (*Harold & Telford, 1990*; *Mooi, 1997*; *Coppard, Zigler & Lessios, 2013*; *Coppard, 2016*; *Coppard & Lessios, 2017*). In addition, twenty-four juvenile *Echinarachnius parma* specimens collected from Lubec, ME, USA were selected for detailed mineralogic analysis using petrographic, powder X-ray diffraction (XRD), scanning electron microscopy (SEM), and EDS methods.

### Analysis using a strong light source

The above-mentioned twenty-four juvenile *E. parma* specimens were first analyzed by projecting an intense light through each individual to confirm the presence and extent of mineral grains inside the diverticulum (Fig. 1A). In addition, selected juvenile *E. parma* specimens were dissected under a stereomicroscope to expose the organ and its contents for subsequent photography (Fig. 1B).

### Optical mineralogy

Selected specimens of *E. parma* were crushed, organic material was removed with sodium hypochlorite, calcite was dissolved with 10% hydrochloric acid, and the remains were washed in alternating baths of distilled water and ethanol. Diverticulum grains from five specimens were mounted on glass slides for optical identification of mineral species using a petrographic microscope.

### Grain size measurements

The size of diverticulum grains was measured directly from SEM micrographs. Substrate samples from Lubec, ME, USA were sieved into size fractions using the sequence of US Sieves 5/10/18/35/60/120/230/pan, which retain grains in the size ranges of ≥4.0 mm/2.0 mm/1.0 mm/0.5 mm/0.25 mm/0.125 mm/0.062 mm/<0.062 mm, respectively.

### X-ray diffraction analysis

Samples for XRD analysis were, depending on species, prepared by crushing 20 to 50 whole specimens after removing organic tissues with sodium hypochlorite. Carbonate material from spines and test was either washed from the samples (which left minor quantities of calcite and, in some cases, aragonite associated with microscopic amounts of sediment retained on the exterior of the specimens), or completely removed with 10% hydrochloric acid. The samples were powdered and mounted in holders for analysis following standard XRD procedures. Bulk samples of substrate sands were dried and powdered without washing or acid treatment. XRD data were acquired using a D5000 diffractometer (Siemens AG, München, Germany).

**Table 1  List of scutelliform specimens incorporated into the present study.**

| Family | Species | Specimen | Locality | Test length | Method | Substrate | Reference |
|---|---|---|---|---|---|---|---|
| Taiwanasteridae | *Sinaechinocyamus mai* | CASIZ 188797 | Tongxiao, Taiwan, 24.49856° N, 120.669374° E | 6.5 mm | µCT | Not available | This study |
| | | NMNS 2689-178 | Off Dingtoue Sandbar, Taiwan, 23.106660° N, 120.036576° E | 7.5 mm | µCT | Not available | This study |
| | | Unvouchered | Tongxiao, Taiwan, 24.489167° N, 120.662500° E | 2–10.5 mm | Optical, X-ray | Available | *Chen & Chen (1994)* |
| Echinarachniidae | *Echinarachnius parma* | Unvouchered | Mowry Beach, Lubec, ME, USA, 44.842678° N, 66.977014° W | 1–25 mm | EDS, Optical, XRD | Available | This study |
| | | Unvouchered | Mowry Beach, Lubec, ME, USA, 44.842678° N, 66.977014° W | 4 mm | µCT | Available | This study |
| | | MCZ Ech-2613 | Off Grand Manan Island, Canada | 10 mm | µCT | Not available | This study |
| | | Unvouchered | Off New Jersey, USA | <10 mm | Optical | Not available | *Serafy (1978), Serafy & Fell (1985)* |
| Scutellidae | *Scaphechinus mirabilis* | ZMB Ech 7405 | Busan, South Korea, 35.150601° N, 129.118146° E | 21 mm | µCT | Not available | This study |
| | | Unvouchered | Grotovaya Bay, Peter the Great Bay, Russia, 42.612401° N, 131.134623° E | Unknown | EDS, Optical | Available | *Elkin et al. (2012)* |
| | | Unvouchered | Grotovaya Bay, Peter the Great Bay, Russia, 42.612401° N, 131.134623° E | Unknown | EDS, Optical | Available | *Elkin et al. (2013)* |
| | | Unvouchered | Troitsa Bay, Peter the Great Bay, Russia, 42.671328° N, 131.114655° E | Unknown | EDS, Optical | Available | *Begun et al. (2014)* |
| Dendrasteridae | *Dendraster excentricus* | CASIZ 094162 | Coos Bay, OR, USA, 43.346120° N, 124.349331° W | 6 and 9 mm | µCT | Not available | This study |
| | | Unvouchered | Alki Point, Seattle, WA, USA, 47.575999° N, 122.420682° W | 5–32 and 70–80 mm | Optical, X-ray | Available | *Chia (1973)* |
| | | Unvouchered | Alki Point, Seattle, WA, USA, 47.575999° N, 122.420682° W; Eastsound, Orcas Island, WA, USA, 48.694258° N, 122.907648° W; False Bay, San Juan Island, WA, USA, 48.491397° N, 123.069311° W | 1.5–30 mm | Optical, X-ray | Not available | *Chia (1985)* |
| | | Unvouchered | Clayton Beach, Bellingham, WA, USA, 48.640429° N, 122.481884° W | Unknown | Optical | Not available | *Mitchell (1972)* |
| Mellitidae | *Encope michelini* | Unvouchered | Grayton Beach, FL, USA, 30.327587° N, 86.167302° W | 12–16 mm | EDS, XRD | Available | This study |
| | | Unvouchered | Grayton Beach, FL, USA, 30.327587° N, 86.167302° W | 14 mm | µCT | Available | This study |
| | | TNSC NPL 4110-4113 | Grayton Beach, FL, USA, 30.327587° N, 86.167302° W | 9–16 mm | µCT | Available | This study |

(Continued)

| Family | Species | Specimen | Locality | Test length | Method | Substrate | Reference |
|---|---|---|---|---|---|---|---|
| | *Encope micropora* | MCZ Ech-2625 | Puntarenas, Costa Rica, 9.974683° N, 84.829808° W | 36 mm | μCT | Not available | This study |
| | *Lanthonia grantii* | USNM E47210 | Punta Mogote, La Paz, Mexico, 24.174844° N, 110.329893° W | 16 mm | μCT | Not available | This study |
| | *Lanthonia longifissa* | Unvouched | Playa Órganos, Paquera, Costa Rica, 9.813242° N, 84.897241° W | 41 mm | X-ray, XRD | Available | This study |
| | *Leodia sexiesperforata* | CASIZ 112813 | Holetown, Barbados, 13.186605° N, 59.638715° W | 5 mm | μCT | Not available | This study |
| | | ZMH E6707 | Puerto Colombia, Colombia, 10.997493° N, 74.955171° W | 8 mm | μCT | Not available | This study |
| | *Mellita notabilis* | Unvouched | Bahia de Banderas, Puerto Vallarta, Mexico, 20.744539° N, 105.429570° W | 27, 30 mm | EDS, XRD | Not available | This study |
| | | Unvouched | Playa Buena Vista, Samara, Costa Rica, 9.879258° N, 85.558507° W | 32 mm | μCT, X-ray | Available | This study |
| | *Mellita quinquiesperforata* | Unvouched | São Sebastião, Brazil, 23.824321° S, 45.396481° W | 2 mm | μCT | Not available | This study |
| | | Unvouched | Matinhos, Brazil, 25.826879° S, 48.534226° W | 1–40 mm | Optical | Available | *Borzone, Tavares & Soares (1997)* |
| | *Mellita tenuis* | Unvouched | Gulf Shores, AL, USA, 30.244595° N, 87.701098° W | 5–25 mm | EDS, XRD | Not available | This study |
| | | Unvouched | Gulf Shores, AL, USA, 30.244595° N, 87.701098° W | 21 mm | μCT | Not available | This study |
| | | CASIZ | Unknown | 6 mm | μCT | Not available | This study |
| | | MCZ Ech-8000 | Marco Beach, FL, USA, 25.932500° N, 81.734029° W | 11 mm | μCT | Not available | This study |
| | *Mellitella stokesii* | USNM E40733 | Playa El Tamarindo, Gulf of Fonseca, El Salvador, 13.193437° N, 87.910424° W | 17 mm | μCT | Not available | This study |

**Note:**
CASIZ, California Academy of Sciences Invertebrate Zoology, San Francisco, CA, USA; EDS, energy-dispersive X-ray spectroscopy; MCZ, Museum of Comparative Zoology, Cambridge, MA, USA; μCT, micro-computed tomography; NMNS, National Museum of Natural Science, Taipei, Taiwan; SEM, scanning electron microscopy; TNSC, Texas Natural Science Center, Austin, TX, USA; USNM, United States National Museum, Washington, DC, USA; XRD, X-ray powder diffraction; ZMB, Museum für Naturkunde, Berlin, Germany; ZMH, Zoologisches Museum Hamburg, Hamburg, Germany.

## Scanning electron microscopy and energy dispersive X-ray spectroscopy

SEM imagery (Fig. 1D) and EDS data were acquired using a JSM-7200 FLV FE-SEM instrument (JEOL, Akishima, Japan). Individual mineral grains from one of the optically analyzed specimens as well as an additional similar specimen were analyzed with EDS for comparison between optical and XRD results. Samples of diverticulum grains were prepared from individual specimens and small splits were mounted on stubs with carbon tape. Some samples were coated with 4–8 nm of platinum, but charging effects were

minimal with grains less than 100 μm in diameter and better results were thus obtained with uncoated grains.

### X-ray imaging
Two-dimensional (2D) X-ray imagery was acquired using a Skyscan 1272 μCT scanning system (Bruker, Kontich, Belgium) in 2D mode.

### Micro-computed tomography
Mineral grains were identified by their X-ray attenuation using μCT at source voltages of 50 to 80 KeV—*Ziegler (2012)* provides parameters for selected specimens. Attenuation of X-rays by minerals was previously described (*Hanna & Ketcham, 2017*), and minerals were identified using μCT based on the MuCalc software (https://www.ctlab.geo.utexas.edu/software/mucalctool/). The variety and general abundance of minerals was constrained by XRD data. MuCalc was used to calculate grain-by-grain mineralogy. Processing of μCT imagery was automated using MATLAB Version 2023a (The MathWorks, Natick, MA, USA). All μCT datasets used in the present study have been deposited in the MorphoBank repository and are available for public download (http://morphobank.org/permalink/?P4915).

### Image processing
Basic image processing was performed using the open source software ImageJ Version 1.52a (https://imagej.net/ij/download.html). In addition, three open source three-dimensional (3D) imaging software packages were used for reslicing and analysis of the μCT scans, *i.e.* SPIERS Version 3.0.1 (https://spiers-software.org/downloads.html), MorphoGraphX Version 1.1.1280 (https://morphographx.org/software/), and SlicerMorph Version 1.4 (https://slicermorph.github.io/), all running on a 64-bit Windows operating system (Microsoft Corp., Redmond, WA, USA).

### Data analysis
Data analysis, including statistical calculations and principal component (PC) analysis, were performed using MATLAB Version 2023a.

### Biogeography
Biogeographic data for scutelliform sand dollar species were obtained from the Global Biodiversity Information Facility (GBIF) database (https://www.gbif.org/).

## RESULTS

### Optical mineralogy, magnetism, and grain size measurements
Results of the optical mineralogic analysis of five selected juvenile specimens of *Echinarachnius parma* are shown in Table 2. Opaque minerals comprise an average of 54.5% of the mineral grains found inside the diverticulum, but cannot be further identified optically. However, opaque minerals are often moderately to strongly magnetic and XRD shows that they comprise magnetite, hematite, and ilmenite. In fact, many diverticulum-bearing sand dollars respond to a strong magnet, a factor directly attributable

**Table 2 Results of the identification of mineral grains found within Gregory's diverticulum in juvenile specimens of *Echinarachnius parma* using optical petrographic methods.**

| Specimen | Test length (mm) | Quartz (light) | Lithic fragments (light) | Clinochlore (light) | Garnet (heavy) | Kyanite (heavy) | Hornblende (heavy) | Augite (heavy) | Zircon (heavy) | Opaque (heavy) | Light minerals | Heavy minerals |
|---|---|---|---|---|---|---|---|---|---|---|---|---|
| EA004 | 4.3 | 6 | 10 | 18 | 4 | 10 | 59 | 103 | 46 | 380 | 5.3% | 94.7% |
| EA007 | 5 | 4 | 10 | 56 | 5 | 26 | 110 | 115 | 190 | 323 | 8.3% | 91.7% |
| EA014 | 5.3 | 1 | 9 | 33 | 2 | 7 | 75 | 66 | 42 | 470 | 6.1% | 93.9% |
| EA002 | 8.6 | 7 | 45 | 34 | 14 | 18 | 10 | 21 | 49 | 175 | 23.1% | 76.9% |
| EA022 | 19.4 | 10 | 26 | 81 | 9 | 13 | 24 | 25 | 138 | 486 | 14.4% | 85.6% |
| Sum | | 28 | 100 | 222 | 34 | 74 | 278 | 330 | 465 | 1,834 | – | – |
| Percentage | | 0.8% | 3.0% | 6.6% | 1.0% | 2.2% | 8.3% | 9.8% | 13.8% | 54.5% | 10.4% | 89.6% |

**Note:**
All specimens collected at Lubec, ME, USA. Light minerals comprise quartz, lithic fragments, and clinochlore. Heavy minerals comprise garnet, kyanite, hornblende, augite, zircon, and opaques (ilmenite as well as magnetite). Values are mineral grain counts and percent of total.

to iron- and titanium-bearing mineral content and also a simple test for the presence of Gregory's diverticulum. However, there is no evidence of magnetotaxis, as juveniles observed in captivity show no preferred orientation with regard to the geomagnetic field (L. Zachos, 2023, personal observation).

The remaining non-opaque minerals inside the above-mentioned specimens comprise primarily zircon, hornblende (an amphibole), and augite (a pyroxene), with minor amounts of other minerals. The light mineral quartz, lithic fragments (comprising in large part quartz and feldspar), and the light mineral clinochlore (chlorite) account for 0.8%, 3.0%, and 6.6%, respectively, of the total, resulting in an average of 89.6% heavy minerals as a remainder (Table 2).

Grain size in the diverticulum is generally in the range of 50 to 150 µm (Fig. 2). This size fraction is very fine to fine sand, which is finer than the fine to medium average grain size of the substrate that these animals inhabit, *i.e.*, 85–90% from 180 to 350 µm. The major mineral composition of grain size fractions of sand substrate from Lubec, ME, USA does not differ substantially from the average composition of bulk samples (Table 3). There is a smaller proportion of free quartz in larger grain size fractions and a corresponding increase in proportion of polymineralic lithic fragments.

### Powder X-ray diffraction analysis

Mineral identification and estimates of overall composition of grains in both specimen and substrate samples were determined using XRD. In every case examined in this study, heavy minerals comprise 70 to >90% of the grains inside the diverticulum, but only 1 to 20% of grains in the substrate (Table 3). Five minerals comprise the major proportion of the heavy mineral component found in Gregory's diverticulum: the iron (Fe) minerals magnetite ($Fe_3O_4$) and hematite ($Fe_2O_3$), the titanium (Ti) minerals ilmenite ($FeTiO_3$) and rutile ($TiO_2$), and the zirconium (Zr) mineral zircon ($ZrSiO_4$). The remaining heavy fraction comprises primarily amphiboles (*e.g.*, hornblende, $Ca_2[(Mg,Fe)_4Al](Al,Si)_7O_{22}(OH)_2$), pyroxenes (*e.g.*, augite, $(Ca,Na)(Mg,Fe,Al,Ti)(Si,Al)_2O_6$), and garnets (*e.g.*, andradite, $Ca_3Fe_2Si_3O_{12}$), with other minor components, and corresponds to the results from optical

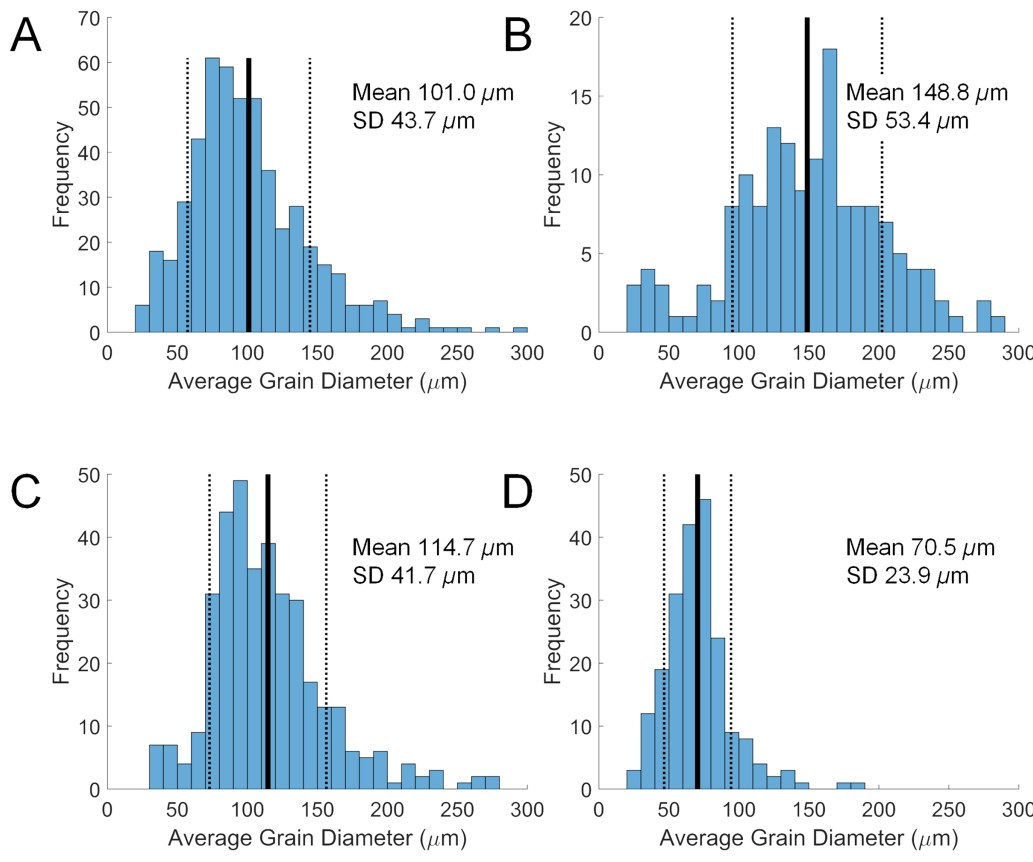

**Figure 2 Size distributions of diverticular grains in four scutelliform species.** (A) *Echinarachnius parma*, (B) *Encope michelini*, (C) *Mellita tenuis*, and (D) *Mellita notabilis*. Measurements were collated from SEM micrographs of grain splits from individual specimens. SD, standard deviation.

**Table 3 Results of X-ray powder diffraction analysis of the mineral content of Gregory's diverticulum and available substrate for selected juvenile scutelliform specimens in comparison with previously published data.**

| Species | Source | Reference | Light minerals (wt %) | Heavy minerals (wt %) | Magnetite (wt %) | Hematite (wt %) | Ilmenite (wt %) | Rutile (wt %) | Zircon (wt %) | Other heavies (wt %) |
|---|---|---|---|---|---|---|---|---|---|---|
| *Sinaechinocyamus mai* | Diverticulum | *Chen & Chen (1994)* | 88.41 | 11.59 | 0 | 11.59 | 0 | 0 | 0 | 0 |
| | Substrate | | 90.07 | 9.93 | 0 | 9.93 | 0 | 0 | 0 | 0 |
| *Echinarachnius parma* | Diverticulum | This study | 27.1 | 72.9 | 0 | 0 | 32.9 | 0 | 19.9 | 20.1 |
| | Substrate | | 97.4 | 2.6 | 0 | 0 | 0 | 0 | 0 | 2.6 |
| *Scaphechinus mirabilis* | Diverticulum | *Elkin et al. (2012)* | 0 | 100 | 0 | 0 | 15 | 0 | 85 | 0 |
| | Substrate | | 88.98 | 11.02 | 0 | 0 | 0.08 | 0 | 0.02 | 10.92 |
| | Diverticulum | *Begun et al. (2014)* | 11.4 | 88.6 | 0 | 0 | 76.2 | 0 | 9.3 | 3.1 |
| | Substrate | | 81.9 | 18.1 | 0 | 0 | 6.9 | 0 | 0.01 | 11.2 |

(Continued)

| Species | Source | Reference | Light minerals (wt %) | Heavy minerals (wt %) | Magnetite (wt %) | Hematite (wt %) | Ilmenite (wt %) | Rutile (wt %) | Zircon (wt %) | Other heavies (wt %) |
|---|---|---|---|---|---|---|---|---|---|---|
| *Dendraster excentricus* | Diverticulum | *Chia (1973)* | 22 | 78 | 78 | 0 | 0 | 0 | 0 | 0 |
| | Substrate | | 90 | 10 | 9.8 | 0.2 | 0 | 0 | 0 | 0 |
| *Encope michelini* | Diverticulum | This study | 20.0 | 80.0 | 0 | 0 | 40.0 | 0 | 40.0 | 0 |
| | Substrate | | 100.0 | 0 | 0 | 0 | 0 | 0 | 0 | 0 |
| *Lanthonia longifissa* | Diverticulum | This study | 23.1 | 76.9 | 20.6 | 18.0 | 20.4 | 0 | 0 | 18.0 |
| | Substrate | | 77.2 | 22.8 | 0 | 0 | 0 | 0 | 0 | 22.78 |
| *Mellita quinquiesperforata* | Diverticulum | *Borzone, Tavares & Soares (1997)* | 55–61 | 45-39 | 19–23 | 0 | 0 | 0 | 0 | 17–25 |
| | Substrate | | 63 | 37 | 4 | 0 | 0 | 0 | 0 | 33 |
| *Mellita notabilis* | Diverticulum | This study | 9.6 | 90.4 | 13.5 | 13.5 | 19.6 | 0 | 43.8 | 0 |
| | Diverticulum | | 15.2 | 84.8 | 59.4 | 3.5 | 19.9 | 0 | 2.1 | 0 |
| | Substrate | | 86.8 | 13.2 | 0 | 0 | 0 | 0 | 0 | 13.2 |
| *Mellita tenuis* | Diverticulum | This study | 23.0 | 77.0 | 0 | 0 | 0 | 16.3 | 60.7 | 0 |

**Note:**
All values are percent of total weight. Heavy minerals are individually tabulated and sum to the percentage of heavy minerals. Published data based on optical methods shown for comparison.

mineralogy (Table 2). However, the exact mix of minerals varies for the different species recovered from different substrates, and the corresponding XRD patterns can be complex.

## Scanning electron microscopy and electron-dispersive spectroscopy analysis

While XRD analysis was here used on samples comprising thousands of mineral grains, SEM and EDS were used to focus on individual grains. The elemental composition of a given sand grain can be simple to complex, and direct correspondence between this grain and a particular mineral species is often impossible when crystallographic information is lacking. However, general patterns emerged from the data when studied using PC analysis to simplify the dataset (Table S1). The first three PCs accounted for 87% of the sample variance. The first (PC 1) was strongly correlated with silicon *vs* titanium and iron content, the second (PC 2) strongly correlated with iron *vs* titanium (Fig. 3A), and the third (PC 3) strongly correlated with aluminum *vs* zirconium (Fig. 3B). The elemental data plotted in the PC space define clusters assignable to broad mineral groups and are consistent with the results obtained from other analytical methods. Point distribution along the first principal axis discriminates the iron and titanium oxides from other oxide, carbonate, and silicate minerals, and distribution along the second principal axis discriminates among magnetite/hematite, ilmenite, and rutile. Point distribution along the third principal axis discriminates zircon and quartz from other silicate minerals. Only two rare earth elements (REE), *i.e.*, Ytterbium (Yb) and Neodymium (Nd) were detected in the samples analyzed, but neither were found at greater than a small fraction of a percent.

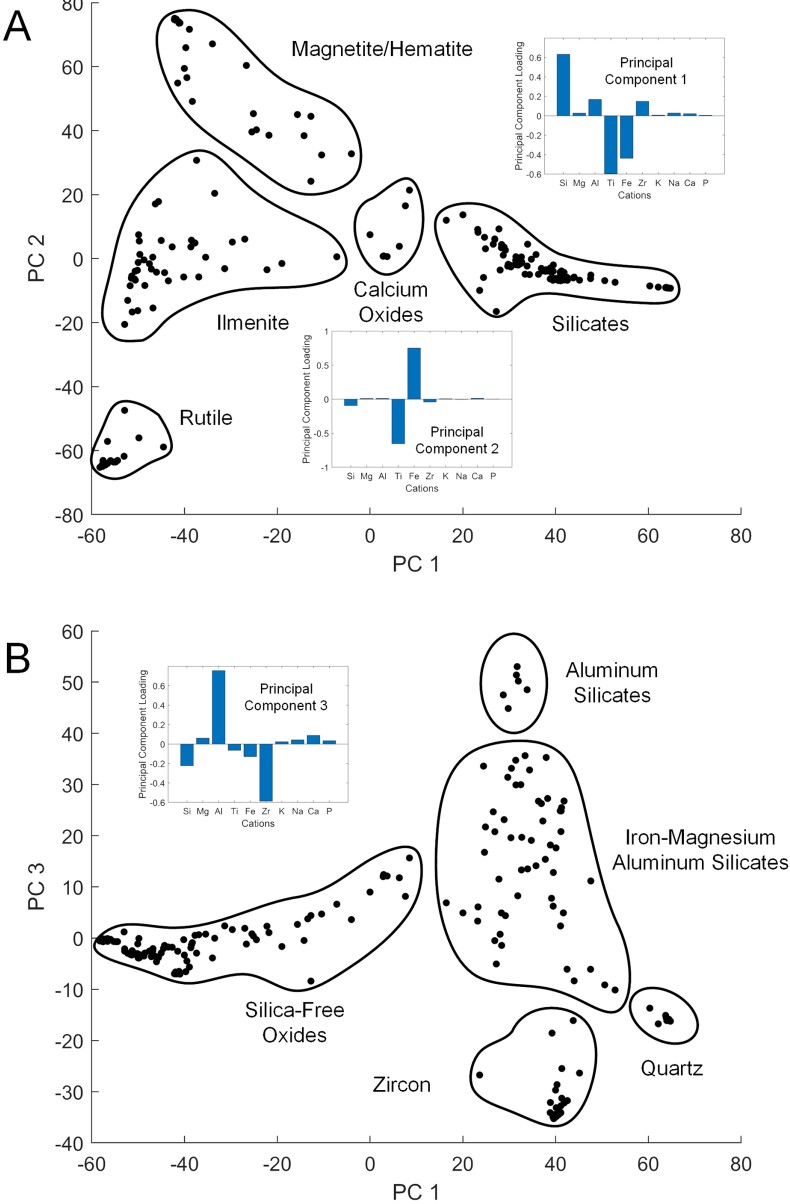

**Figure 3 Principal component cross-plots showing point clusters associated with minerals or mineral classes.** Data are from EDS analysis of mineral grains. Subplots show principal component (PC) loadings in terms of major cations. (A) PC 1 *vs* PC 2. PC 1 loadings emphasize iron and titanium *vs* silicon, with heavier minerals in the negative range. PC 2 loadings emphasize iron *vs* titanium, differentiating the primary metal oxide minerals. (B) PC 1 *vs* PC 3. PC 3 loadings emphasize zirconium *vs* aluminum, with heavier minerals in the negative range.

## Micro-computed tomography analysis

Use of μCT has the advantage of permitting identification of grain mineralogy and, by automated counting of the individual grains, estimation of the proportion of different minerals. The analyzed mineral composition from Gregory's diverticulum found in twelve scutelliform species conforms, in general, with results derived from optical, XRD, and EDS analyses (Table 4). Because of similarity in X-ray attenuation, magnetite and hematite

**Table 4 Results of micro-computed tomography analysis of the mineral content of Gregory's diverticulum in selected juvenile scutelliform specimens.**

| Specimen | Quartz (grain %) | Calcite (grain %) | Hornblende (grain %) | Garnet (grain %) | Magnetite & Hematite (grain %) | Ilmenite (grain %) | Rutile (grain %) | Zircon (grain %) | Heavy Minerals (grain %) |
|---|---|---|---|---|---|---|---|---|---|
| *Sinaechinocyamus mai* CASIZ 118797, 6.5 mm | 1.55 | 0.00 | 0.62 | 0.00 | 24.84 | 4.04 | 1.55 | 67.39 | 98.44 |
| *Sinaechinocyamus mai* NMNS 2689-178, 7.5 mm | 0.16 | 0.00 | 0.00 | 0.00 | 0.00 | 0.00 | 0.00 | 99.84 | 99.84 |
| *Echinarachnius parma* Unvouchered, 4 mm, virtual section 200 | 1.56 | 0.39 | 0.00 | 0.39 | 38.13 | 54.86 | 0.00 | 4.67 | 98.05 |
| *Echinarachnius parma* Unvouchered, 4 mm, virtual section 225 | 0.00 | 0.00 | 0.00 | 13.09 | 41.36 | 41.01 | 0.00 | 4.54 | 100.00 |
| *Echinarachnius parma* Unvouchered, 4 mm, virtual section 250 | 1.89 | 0.17 | 0.00 | 2.06 | 53.17 | 36.02 | 0.00 | 6.69 | 97.94 |
| *Echinarachnius parma* Unvouchered, 4 mm, virtual section 275 | 2.27 | 0.00 | 1.76 | 2.52 | 61.96 | 22.17 | 0.00 | 8.31 | 97.73 |
| *Echinarachnius parma* MCZ Ech-2613, 10 mm | 0.00 | 0.27 | 1.57 | 5.57 | 33.41 | 23.75 | 0.00 | 35.43 | 99.73 |
| *Scaphechinus mirabilis* ZMB Ech 7405, 21 mm | 0.04 | 0.31 | 0.19 | 0.00 | 64.08 | 24.43 | 2.75 | 8.20 | 99.65 |
| *Dendraster excentricus* CASIZ 094162, 6 mm | 0.72 | 0.72 | 1.56 | 0.00 | 39.95 | 17.70 | 2.39 | 36.96 | 98.56 |
| *Dendraster excentricus* CASIZ 094162, 9 mm | 2.24 | 3.16 | 8.68 | 0.00 | 12.76 | 26.58 | 30.66 | 15.92 | 94.60 |
| *Encope michelini* TNSC NPL 4110, 9 mm | 3.99 | 3.26 | 5.8 | 0.00 | 54.35 | 17.39 | 9.42 | 5.8 | 92.76 |
| *Encope michelini* TNSC NPL 4111, 9 mm | 0.00 | 12.56 | 18.36 | 0.00 | 34.78 | 15.94 | 14.98 | 3.38 | 87.44 |
| *Encope michelini* TNSC NPL 4112, 13 mm | 0.00 | 19.17 | 19.17 | 0.00 | 20.83 | 18.33 | 21.67 | 0.83 | 80.83 |
| *Encope michelini* Unvouchered, 14 mm | 20.95 | 10.60 | 10.35 | 0.00 | 17.30 | 12.90 | 7.32 | 20.58 | 68.45 |
| *Encope michelini* TNSC NPL 4113, 16 mm | 2.15 | 10.75 | 13.98 | 0.00 | 31.18 | 13.98 | 22.58 | 5.38 | 87.1 |
| *Encope micropora* MCZ Ech-2625, 36 mm | 0.13 | 0.11 | 0.15 | 0.00 | 79.86 | 8.20 | 0.65 | 10.89 | 99.75 |
| *Lanthonia grantii* USNM E47210, 16 mm | 0.00 | 0.05 | 4.07 | 0.00 | 63.73 | 14.11 | 6.51 | 11.53 | 99.95 |
| *Leodia sexiesperforata* CASIZ 112813, 5 mm | 4.27 | 3.35 | 1.98 | 0.00 | 52.13 | 11.59 | 4.88 | 21.8 | 92.38 |
| *Leodia sexiesperforata* ZMH E6707, 8 mm | 5.30 | 0.66 | 2.65 | 0.00 | 48.34 | 26.49 | 13.91 | 2.65 | 94.04 |
| *Mellita notabilis* Unvouchered, 32 mm | 0.01 | 0.00 | 0.28 | 0.00 | 65.12 | 29.21 | 5.13 | 0.25 | 99.99 |

| Table 4 (continued) | | | | | | | | | |
|---|---|---|---|---|---|---|---|---|---|
| Specimen | Quartz (grain %) | Calcite (grain %) | Hornblende (grain %) | Garnet (grain %) | Magnetite & Hematite (grain %) | Ilmenite (grain %) | Rutile (grain %) | Zircon (grain %) | Heavy Minerals (grain %) |
| *Mellita quinquiesperforata* Unvouchered, 2 mm | 0.97 | 0.00 | 0.48 | 0.00 | 33.82 | 59.90 | 4.83 | 0.00 | 99.03 |
| *Mellita tenuis* CASIZ, 6 mm | 0.88 | 0.44 | 5.31 | 0.00 | 50.88 | 11.06 | 4.42 | 26.99 | 98.66 |
| *Mellita tenuis* Unvouchered, 21 mm | 0.47 | 0.57 | 0.00 | 0.00 | 10.41 | 4.45 | 0.47 | 83.63 | 98.96 |
| *Mellitella stokesii* USNM E40733, 17 mm | 0.59 | 0.79 | 1.58 | 0.00 | 6.01 | 75.76 | 15.27 | 0.00 | 98.62 |

**Note:**

Mineral content of representative specimens with specimen ID and test length. Data include mineral content from separate µCT. virtual sections, approximately 225 *µm* apart, from a single specimen of *Echinarachnius parma* with 4 mm test length from Lubec, ME, USA. All values are in percent of total grains. See Table 1 for additional information on each specimen.

cannot be differentiated and are thus here combined. Regardless of the species or its origin, the proportion of heavy minerals exceeded 68%, with ilmenite, magnetite/hematite, rutile, and zircon making up the largest proportion of the heavy mineral fraction (Table 4). Analyses of individual specimens as well as multiple virtual µCT sections through the same specimen of *E. parma* (Table 4) demonstrate variation in the proportions of different minerals, but the mix of heavy minerals in all cases exceeded 90% of the total grains for this species.

# DISCUSSION

## Methodological approach

The identification and description of the mineral grains retained in Gregory's diverticulum can be accomplished using a variety of methods and while each of these generates similar results, they are not equivalent in either application or interpretation. The most precise measurements involve analysis of individual grains, which themselves may be mono- or polymineralic. However, these methods (optical, XRD, SEM and EDS) require removal of grains from the specimen and are therefore inherently destructive. In contrast, non-destructive methods such as X-ray or µCT rely on an attenuation proxy that does not directly indicate the mineralogy of a mineral grain. However, given sufficiently large sample sizes, the results of the destructive methods can be combined with non-destructively obtained data to develop a general method for estimating the type, variation, and relative abundance of the various minerals across multiple species. Our research objective with regard to methodology was therefore two-fold: 1) choose species for which large numbers of specimens are available for destructive as well as non-destructive analysis, and then 2) extend these results to those species for which only a limited number of specimens is available or which could not be sacrificed using destructive methods.

The relatively common sand dollar species *Echinarachnius parma* was here selected to represent the model species. Hundreds of juvenile specimens with a test length <20 mm,

collected from the Bay of Fundy at Lubec, ME, USA, were available for both destructive and non-destructive analyses. Two simple tests can be used to determine whether or not a specimen contains heavy minerals. 1) The specimens of *E. parma* from Lubec, ME, USA contain enough iron and titanium minerals to be susceptible to a magnet, and this was also the case with several other species tested. 2) Additionally, a strong light source can be focused through a specimen to reveal a silhouette of the mineral-filled diverticulum (Fig. 1A). The specimen can then be dissected (or crushed) and a quick estimate of the relative proportion of light to heavy minerals can be made by separating the opaque grains from the transparent grains (Figs. 1B, 1C and 1E). In case of *E. parma*, many of the remaining mineral grains are often zircon, which can be recognized by the distinctive prismatic euhedral shape of the grains and by the strong orange-yellow fluorescence when illuminated with short wavelength (254 nm) ultraviolet (UV) light. Identification of individual grains using a petrographic microscope is restricted to the non-opaque fraction of the total grain content and limited by the time-consuming nature of the process. Nonetheless, its primary utility is in identification of the relatively common amphiboles, pyroxenes, and garnets as well as some of the other minor fractions of mineral grains. In turn, XRD is useful for its definitive identification and quantitative measures of the major mineral fractions derived from the diverticulum grains. However, a major drawback is the amount of material needed for a valid analysis, usually requiring the extraction of grains from 20 to 50 specimens. The XRD patterns with a mix of mineralogy exhibiting overlapping diffraction peaks complicate strict quantitative measures of composition, but are invaluable for the qualitative interpretation of the overall mineralogy and relative abundance of the individual mineral species. Unfortunately, combined SEM/EDS analysis suffers from one of the same drawbacks as the petrographic method, *i.e.*, the time-consuming focus on individual grains, although the process is more automated and is not restricted by grain opacity. The elemental data derived from the EDS are invaluable and can usually be used to identify either the mineralogy of the sampled grain or at least the major mineral group to which it belongs.

Minerals attenuate X-rays in accordance with their elemental composition and crystal structure, and therefore attenuation can be employed as a proxy for mineralogy. Simple 2D X-ray imaging is an efficient non-destructive method to display the grain-filled diverticulum in a specimen, but is of limited value in grain identification. However, μCT imaging is an effective method for non-destructively quantifying the mineral content of a specimen in 3D. In the present study, a single virtual horizontal section was selected from each μCT 16-bit image stack, for which a pixel value of 0 indicates no attenuation and a pixel value of 65,535 (*i.e.*, $2^{16}-1$) indicates 100% attenuation. This virtual section was processed to calculate the local intensity maxima, which approximate the dense centers of the grains (Fig. 4). The resolution of the maxima is dependent on the pixel resolution of the image (*i.e.*, on the isotropic voxel resolution of the μCT scan). Low-resolution images only resolve a generalized density map that approximates grain centers, each representing a single local intensity maximum. The relative attenuation, as measured by image intensity, was scaled between the known absolute attenuation value for calcite and zircon. If no zircon was present or if the image gain was too high and the zircon signal was saturated,
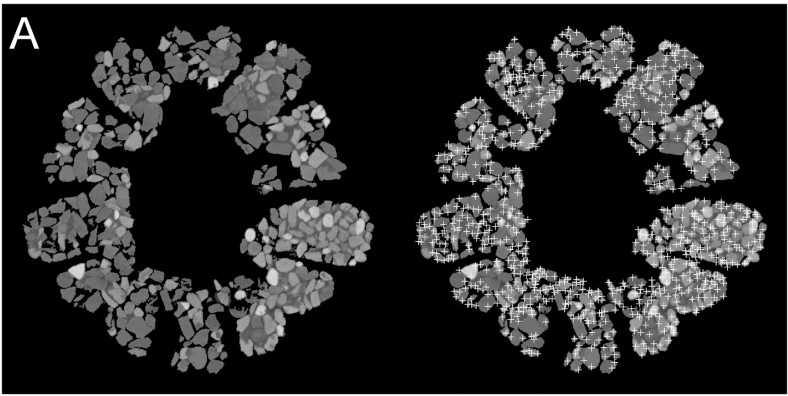

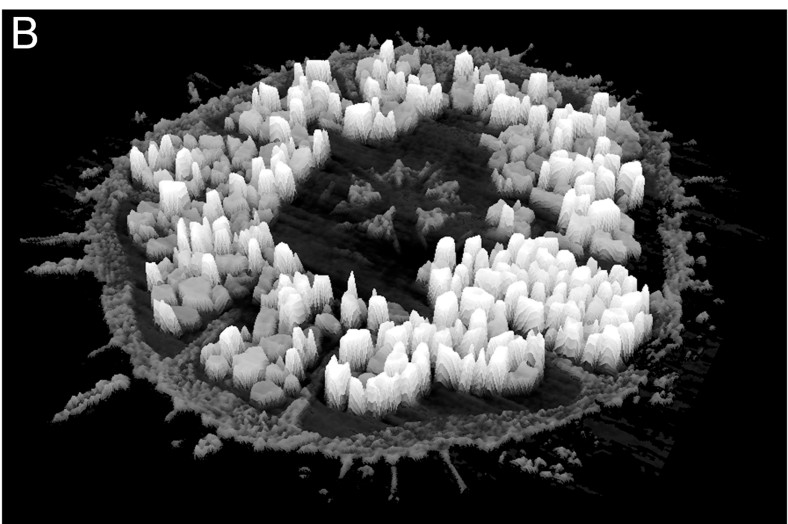

**Figure 4 Examples of automatic grain sampling virtual sections of a micro-computed tomography dataset of *Echinarachnius parma*.** (A) Montage showing on the left a filtered image of mineral grains inside Gregory's diverticulum and on the right an overlay of sampling points that mark local maxima of pixel values—these approximate the points of maximum X-ray attenuation for individual grains. (B) Perspective rendering of a selected μCT virtual section, where the pixel values are extruded as height above the background. The calcite of the test can be readily distinguished from the heavy minerals inside Gregory's diverticulum.

then magnetite, the mineral with the next highest attenuation coefficient, was used as reference. This scaling resulted in the 16-bit image intensity values that could be expected for each of the analyzed minerals. Low-intensity peaks representative of the calcite skeleton were removed with a high-pass filter and the remaining mean intensities were compared with the expected values generated from the software MuCalc to estimate both the identity and relative volume percent of each mineral (see Figs. S1 and S2 for exemplary X-ray images and extracted intensity distributions). Workable solutions were also derived from 8-bit images upsampled to 16-bit, but here the significant loss of spectral resolution is undesirable, particularly in case of low pixel resolution.

However, one difficulty with mineral determination from μCT imagery is the thickness variation in free mineral grains. The relationship between total absorption and the

coefficient of attenuation can be represented by the Lambert-Beer Law (*Hanna & Ketcham, 2017*):

$$\frac{I}{I_0} = e^{(-\mu x)}$$

where *I* is the final X-ray intensity, $I_0$ is the initial X-ray intensity, $\mu$ is the linear attenuation coefficient, and *x* is the path length through the grain. The maximum reduction in X-ray intensity is at the center of the grain, but a distribution of intensities across a grain will have a peak related to the average thickness. The virtual sections in a µCT image sequence have a constant thickness, but measured intensity values are dependent on X-ray transmission through a sample in 3D and are susceptible to beam-hardening effects. The probability density functions used to model the intensity peaks are broadened and skewed around the means by these non-linear effects, adding uncertainty to quantitative interpretation.

Nonetheless, the advantages of the approach developed for the present study include the non-destructive nature of the analysis, calculation of mineral distributions directly from grain populations of individual specimens, and the ability to automate the analysis computationally. One example for the usefulness of the approach introduced here is the case of the previously mentioned species *Sinaechinocyamus mai*—data on diverticulum grains obtained using optical mineralogy suggested approximately the same proportion of light to heavy minerals as found in the substrate (*Chen & Chen, 1994*). Instead, our calibrated and standardized µCT data strongly suggest the presence of ca. 25% magnetite and 67% zircon in the first and almost 100% zircon in the second specimen analyzed (Fig. S2F and Table 4). Unfortunately, while X-ray attenuation is directly related to the density of a mineral, it is often equivocal with regard to actual mineral identification. Where the X-ray attenuation of minerals is similar, such as for almandine and rutile or for magnetite and hematite, accurate mineral identification relies on knowledge of the expected mineral population of the substrate. In many cases, this information is available from local or regional mineralogical studies of nearshore sediments, or can be inferred from the geologic character of the source rocks.

## Mineral selection

Our results demonstrate that five minerals account for most of the heavy mineral fraction of the scutelliform species analyzed, *i.e.*, magnetite ($Fe_3O_4$), hematite ($Fe_2O_3$), ilmenite ($FeTiO_3$), rutile ($TiO_2$), and zircon ($ZrSiO_4$). Minor heavy minerals include various amphiboles, pyroxenes, and garnets generally at concentrations too small to identify reliably. Derived sand dollars with Gregory's diverticulum significantly concentrate heavy minerals when compared to the composition of the substrate. A two-sample *t*-test using the data provided in Table 3 rejected the null hypothesis that the mean ratios of light to heavy minerals between the mineral grains of diverticulum and substrate are drawn from the same population, with a *p*-value $< 0.05$. However, the particular mix of mineralogy for any given species appears to be more a function of the heavy mineral composition of the substrate than any particular mineral selectivity on the part of the animal. For example,

**Table 5 Physical properties of primary light and heavy minerals found inside Gregory's diverticulum.**

| Mineral | Density (g/cm³) | Chemical formula | Magnetism | Conductivity | X-ray linear attenuation μ coefficient (60 KeVcm⁻¹) | Wetability | Contact angle (°) | Zeta Potential (at pH 8) (mV) |
|---------|------|-------|----------|---------------|-----------|-----------|-------|--------|
| Albite | 2.62 | $NaAlSi_3O_8$ | Diamagnetic | Non-conductive | 0.245[3] | Hydrophilic | 0.0[1] | −31.38[4] |
| Quartz | 2.65 | $SiO_2$ | Diamagnetic | Non-conductive | 0.251[3] | Hydrophilic | 0.0[1] | −60.00[7] |
| Rutile | 4.23 | $TiO_2$ | Diamagnetic | Conductive | 0.536[3] | Hydrophilic | 0.0[1] | −28.82[5] |
| Zircon | 4.6-4.7 | $ZrSiO_4$ | Nonmagnetic | Non-conductive | 1.980[3] | Hydrophilic | 0.0[1] | −31.37[5] |
| Ilmenite | 4.70–4.79 | $FeTiO_3$ | Paramagnetic | Conductive | 0.746[3] | Hydrophobic | 14.0[1] | −28.25[5] |
| Magnetite | 5.15 | $Fe_3O_4$ | Ferromagnetic | Conductive | 0.925[3] | Hydrophobic | 34.16[8] | −35.14[2] |
| Hematite | 5.26 | $Fe_2O_3$ | Ferromagnetic | Conductive | 0.900[3] | Hydrophobic | 32.67[8] | −26.71[6] |

**Notes:**
Source of data for X-ray attenuation coefficients, contact angles, and zeta potential values are given in footnotes. Contact angle is distilled water-air interface and zeta potential is the electrical potential across the mineral-water interface at pH 8 of seawater, adjusted in distilled water with KOH. These values will vary with solute concentration of the liquid, liquid temperature, and pH and do not represent conditions in the scutelliform main digestive tract or in Gregory's diverticulum.

[1] *Drzymała (2007)*.
[2] *Erdemoğlu & Sarikaya (2006)*.
[3] *Hanna & Ketcham (2017)*.
[4] *Kursun (2010)*.
[5] *Nduwa-Mushidi (2016)*.
[6] *Quast (2006)*.
[7] *Ruan et al. (2018)*.
[8] *Tang et al. (2018)*.

specimens of *Mellita notabilis* collected from Bahia de Banderas, Mexico contain more zircon than specimens of *M. notabilis* collected from Playa Buena Vista, Costa Rica (Table 3). A similar variability was documented for *Scaphechinus mirabilis* (*Elkin et al., 2012*; *Begun et al., 2014*; *Elkin, 2019*). In turn, different specimens of *E. parma* from Lubec, ME, USA contain the same minerals, but in varying percentages (Tables 2 and 4). Exactly how the animals differentiate heavy from light minerals that are both present in the substrate is not known, but there are two likely methods: (1) direct sensing of the specific gravity (or heft) of a mineral grain or (2) sensing of an electrochemical surface character of the mineral grain–so, essentially by touch or by taste. The mystery surrounding the mechanism whereby these animals select and retain heavy minerals is deepened by the consideration that some agglutinated foraminifera also select and retain heavy minerals in their cytoplasm or even test (*Makled & Langer, 2010*; *Sabbatini et al., 2016*; *Garrison, 2019*). These single-celled organisms select primarily the heavy minerals zircon, rutile, and ilmenite with a mean grain size of about 100 μm, although *Waśkowska (2014)* described a fossil species that preferentially selected tourmaline with a grain size of 39 to 69 μm (predominantly dravite, with a specific gravity of 3.0–3.2 g/cm³).

Minerals vary considerably in wettability (*Ozcan, 1992*), contact angle (*Drzymała, 2007*; *Tang et al., 2018*), magnetic susceptibility (*Rosenblum & Brownfield, 2000*), and zeta potential (*Erdemoğlu & Sarikaya, 2006*; *Quast, 2006*; *Kursun, 2010*; *Nduwa-Mushidi, 2016*; *Nduwa-Mushidi & Anderson, 2017*; *Ruan et al., 2018*). Taken together, these physical properties could indicate commonalities among the primary minerals found inside Gregory's diverticulum (Table 5), which in turn could play a role in selectivity. However, while it has previously been hypothesized that sand dollars select minerals based on REE content, particularly in zircon mineral grains (*Panichev, 2015*), REE content of the

minerals analyzed here was found to be insignificant, although our EDS analyses are incomplete with regard to the full range of species retaining heavy minerals.

Some size selectivity for grains obtained from the substrate is observed, with a clear preference for grains in the 50 to 150 μm range (Fig. 2). Observations on living *E. parma* demonstrate that, for the smallest juveniles (1 to 5 mm test length), the diameter of the periproct (*i.e.*, the anal opening through the test) limits the ability to pass grains larger than 150 μm out of the rectum. Interestingly, the size of diverticulum grains does not change as the animal grows, whereas the diameter of the periproct increases with age and size. Thus, while the upper limit of grain size appears to be constrained by the diameter of the periproct and the maximum opening of the teeth in the smallest juveniles, it may also be constrained by the diameter of the duct connector throughout ontogeny (Fig. 1C). If the limiting factor is indeed the duct connector then this would imply that grain selection occurs internally and is likely a passive process, whereas grain selection before ingestion implies active selection by the tube feet and/or spines. For example, stereomicroscopic studies on living *E. parma* demonstrate that heavy mineral grains can be selected by tube feet on the oral surface of the animal, then transported to the ambulacral grooves and finally—with the aid of the spines—into the peristome (*i.e.* the oral opening) as shown in Video S1. The buccal tube feet, in conjunction with the oral spines, actively select and incorporate (Video S2) or reject and discard (Video S3) mineral grains, supporting the hypothesis that selection of heavy minerals occurs before the grains are ingested (*Chia, 1985*). However, the precise means by which the tube feet and spines distinguish between individual mineral grains remains unknown at this point.

The mean grain size of the substrate is larger than that of the diverticulum grains and is 85 to 90% in the range of 180 to 350 μm, or fine to medium sand. This is in agreement with previous studies on substrate preference of *E. parma* (*Stanley & James, 1971*; *Serafy, 1978*; *Harold & Telford, 1982*; *Serafy & Fell, 1985*), although *Brown (1983)* reported a preference for coarse sand. *Mellita tenuis* shows a similar preference for fine to medium substrate grain size (*Pomory, Robbins & Lares, 1995*), as do *Leodia sexiesperforata*, *Mellita quinquiesperforata*, and *Encope michelini* (*Weihe & Gray, 1968*; *Telford & Mooi, 1986*; *Hilber & Lawrence, 2009*). While it has previously been claimed that *L. sexiesperforata* inhabits only biogenic carbonate substrates (*Telford & Mooi, 1986*), a strict preference for biogenic sands has not been verified (*Mooi & Peterson, 2000*). While we did not have access to substrate samples for the *L. sexiesperforata* specimens analyzed in this study, the bioclastic beach sands on the western side of Barbados, where one specimen was collected (Table 1), can contain up to 10% combined quartz and heavy mineral components (*Limonta et al., 2015*).

The observation that the concentration of light minerals (primarily quartz and feldspar) can range from 0% to 25% of diverticulum grains and that also minor amounts of other minerals may occur provides evidence that sometimes the animals simply pick up what is readily available in the substrate. Sedimentologic studies distinguish shallow-water coastal provinces characterized by specific heavy mineral suites. In the western hemisphere, these studies are limited in scope to the Atlantic coast of North America (*Van Gosen & Ellefsen, 2018*), dominated by ilmenite and zircon; the Gulf of Mexico (*Davis, 2017*), dominated by

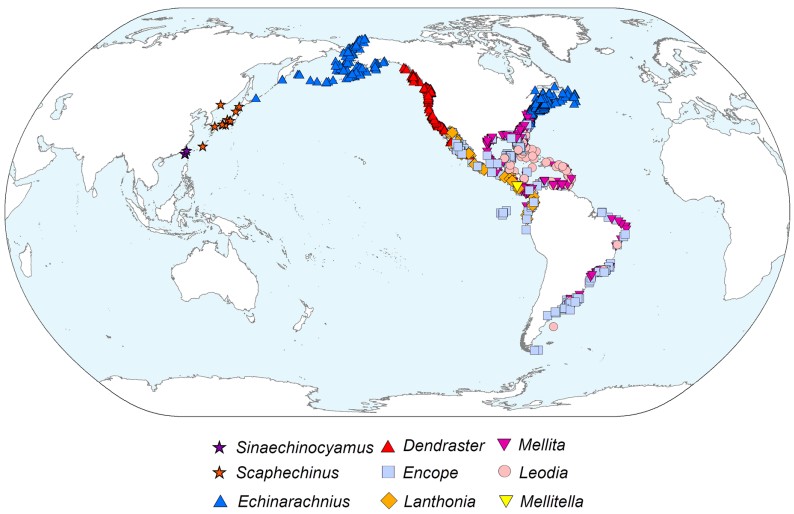

★ *Sinaechinocyamus*  ▲ *Dendraster*  ▼ *Mellita*
★ *Scaphechinus*  ▪ *Encope*  ● *Leodia*
▲ *Echinarachnius*  ◆ *Lanthonia*  ▽ *Mellitella*

**Figure 5 Geographic distribution of representatives from all nine extant scutelliform genera.** Occurrence data coordinates downloaded from open-source Global Biodiversity Information Network (https://www.gbif.org) with the following links. *Sinaechinocyamus*: GBIF.org (24 May 2019) GBIF Occurrence Download https://doi.org/10.15468/dl.hurvee; *Scaphechinus*: GBIF.org (24 May 2019) GBIF Occurrence Download https://doi.org/10.15468/dl.kky6kn; *Echinarachnius*: GBIF.org (04 March 2021) GBIF Occurrence Download https://doi.org/10.15468/dl.zam5tz; *Dendraster*: GBIF.org (24 May 2019) GBIF Occurrence Download https://doi.org/10.15468/dl.scpqbt; *Encope*: GBIF.org (04 March 2021) GBIF Occurrence Download https://doi.org/10.15468/dl.btrwe6; *Lanthonia*: GBIF.org (24 May 2019) GBIF Occurrence Download https://doi.org/10.15468/dl.mz33pi; *Mellita*: GBIF.org (04 March 2021) GBIF Occurrence Download https://doi.org/10.15468/dl.3s8sg2; *Leodia*: GBIF.org (04 March 2021) GBIF Occurrence Download https://doi.org/10.15468/dl.u4x7m5; *Mellitella*: GBIF.org (24 May 2019) GBIF Occurrence Download https://doi.org/10.15468/dl.futtrn. Made with Natural Earth. Free vector and raster map data @ naturalearthdata.com.

rutile and zircon; and the Pacific coast of Mexico (*Carranza-Edwards et al., 2009, 2019*), dominated by magnetite and zircon. Heavy minerals in the Okhotsk Sea are predominantly hornblende, hypersthene, and epidote, with only minor amounts of ilmenite, magnetite, and zircon (*Wang et al., 2021*). In the southern Sea of Japan, shore sediments are predominantly hypersthene and hornblende, with only minor amounts of zircon (*Yokota et al., 1990*). Heavy minerals in the lower reaches of rivers in western Taiwan are predominantly zircon and garnet (*Deng et al., 2016*; *Garzanti et al., 2023*). In addition, while the overall heavy mineral concentration of the substrate can be determined and compared with the concentration inside the diverticulum, it is also possible that the animals selected mineral grains from a naturally concentrated source. For example, heavy minerals in littoral sands are usually deposited in thin layers or "stringers" with much higher concentration than the overall deposit (*Van Gosen & Ellefsen, 2018*).

## Phylogeography

The diverticulum-bearing scutelliforms are restricted to the eastern and western coasts of North and South America as well as the northeastern coast of Asia (Fig. 5). The Astriclypeidae, a scutelliform taxon hypothesized to be sister to all other extant

scutelliforms (*Lee et al., 2023*) is not found in these regions (*Ghiold & Hoffman, 1986*) and lacks Gregory's diverticulum (*Ziegler et al., 2016*). The oldest fossil records of scutelliform echinoids (Eoscutellidae and Protoscutellidae) occur in North America, and it has long been proposed that this clade originated there (*Stefanini, 1924*). Strictly European fossil scutelliform genera (*Scutella, Parascutella, Remondella, Scutulum, Samlandaster*) were distinguished by *Durham (1955)*, and no evidence of Gregory's diverticulum has been reported for any of these. Such lack of evidence is supported by a recent study of juvenile specimens from several protoscutellid species from Eocene deposits of eastern North America that has failed to produce definitive evidence for the presence of a mineral-filled diverticulum (*Zachos & Ziegler, 2021*). To date, the oldest proven occurrence of a mineral-filled diverticulum is therefore that of *Kewia marquamensis* (Echinarachniidae) from the late Oligocene of Oregon (*Linder, 1986*; *Linder, Durham & Orr, 1988*), suggesting that the lineage of scutelliforms with the organ evolved in western North America in the middle to late Paleogene.

## CONCLUSIONS

Scutelliform sand dollars that possess Gregory's diverticulum fill this internal organ with mineral grains in the very fine to fine sand size range. In every case, the proportion of heavy to light minerals greatly exceeds that of the substrate which the animal inhabits, with heavy mineral content ranging from ~70 to >90% of the mineral grains retained. Minerals containing iron (magnetite and hematite), titanium (ilmenite and rutile), and zirconium (zircon) represent the major portion of heavy minerals found inside the organ.

Several different methods can be used to analyse the mineral content of Gregory's diverticulum. Magnetite and ilmenite grains often make a specimen magnetic, a simple test for the presence of the organ. Other non-destructive methods include using a strong light source or X-ray imaging. Destructive methods such as optical mineralogy, XRD or SEM/EDS require extraction of the mineral grains through dissection or crushing, but lead to precise mineral or elemental identification. The non-destructive process using µCT proposed here can be employed to closely estimate light to heavy mineral proportions and generate a reliably accurate estimation of actual mineral species composition.

The intra- and inter-species variation in heavy mineral content leads to the conclusion that the selection of a particular mineral grain is opportunistic, in the sense of exploiting chances offered by immediate circumstances. Any mineral grain of the proper size is acceptable to the animal as long as it has a high specific gravity, with a strong preference for density exceeding 4 $g/cm^3$. The apparent correlation of heavy mineral composition of the substrate and the choice of heavy minerals in the diverticulum further supports the contention of opportunism. The restriction of minor component grains to densities between 2.9 to 4.0 $g/cm^3$, rejecting nearly all lighter minerals such as quartz and calcite, suggests that grain choice is based on the weight of the grains, but does not rule out other selection mechanisms. The upper limit on grain size appears to be constrained by the

diameter of the periproct and peristome. The size of diverticulum grains does not increase with the size of the animal, which could be the result of size selectivity of grains during consumption or could also be caused by the limiting diameter of the duct connector. The selectivity for density observed in all analyzed species is further evidence that Gregory's diverticulum does indeed serve as a weight belt and may have an adaptive value for the hydrodynamic stability of juveniles in a high-energy fluid regime (*Chia, 1973*), although there is no observational or experimental evidence to support this (*Chen & Chen, 1994*). Whether or not this is the only adaptive value for a grain-filled diverticulum is still undetermined (*Lawrence, 2001*).

The presence of Gregory's diverticulum appears to be phylogenetically informative, and the fossil record suggests that the common ancestor of all diverticulum-bearing scutelliforms evolved during the late Paleogene and, considering the current biogeography, likely originated in the northeastern Pacific along the western coast of North America. This clade could have then dispersed along the coast southward to South America and into the Caribbean (prior to the closure of the Isthmus of Panama) as well as north- and then later westward to Asia and possibly also eastward through boreal waters along the northwestern Atlantic coast (*Ghiold & Hoffman, 1986*).

There is no definitively known mechanism whereby these organisms can select heavy over light mineral grains, nor why they have a preference for iron-, titanium-, and zirconium-bearing mineral grains—other than the fact that these are among the heaviest of heavy minerals. Likewise, there is no clear reason as to why grains are stored in the diverticulum in juveniles, but in most (although not all) species expelled as the organism matures, followed by atrophy of the diverticulum tissue. However, there seems to be little question that these animals have developed the behaviour of actively selecting mineral grains with not only high specific gravity, but also specific composition and size, leading to the conjecture that the behaviour preceded the development of the diverticulum itself. The apparent persistence of this trait since the Oligocene and its ubiquitous expression within a clearly delineated clade of derived sand dollars across tropical to boreal environments is evidence that it is highly adaptive, yet it remains as of yet unclear exactly what the adaptive advantage might be.

## ACKNOWLEDGEMENTS

We would like to thank Thomas Bartolomaeus and Thorsten Geisler-Wierwille for hospitality and use of facilities; Harald Euler and Hans Henning Friedrich for their help in conducting XRD analyses; Vijayasankar Raman for help in conducting EDS analyses; Jennifer Gifford for help with optical mineralogy; Jennifer W. Trimble for help with μCT scanning; Janik Bollé for carrying out dissections and SEM; and Juan José Alvarado Barrientos, Francisco A. Solís Marín as well as Alvaro E. Migotto for help with specimen collection. We are grateful to the curators of the museum collections accessed for this study. We also thank Anna G. Kral, Stuart R. Stock, and Andreas Ziegler for helpful discussions and comments on the manuscript.

### Funding

SEM analyses presented in this work were conducted at the Microscopy and Imaging Center (SEM Core) of the University of Mississippi. This facility was supported by grant 1726880, National Science Foundation. μCT scanning obtained from the Bonner Institut für Organismische Biologie at Rheinische Friedrich-Wilhelms-Universität was supported by DFG grant INST 217/849-1FUGG. μCT scans obtained from the High-Resolution X-ray Computed Tomography Facility at The University of Texas at Austin were supported by the NSF Division of Earth Science Instrumentation and Facilities Program (NSF EAR-1762458) and NASA (80NSSC23K0199). There was no additional external funding received for this study. The funders had no role in study design, data collection and analysis, decision to publish, or preparation of the manuscript.

### Grant Disclosures

The following grant information was disclosed by the authors:
University of Mississippi.
National Science Foundation: 1726880.
DFG: INST 217/849-1FUGG.
NSF Division of Earth Science Instrumentation and Facilities Program: NSF EAR-1762458.
NASA: 80NSSC23K0199.

### Competing Interests

The authors declare that they have no competing interests.

### Author Contributions

- Louis G. Zachos conceived and designed the experiments, performed the experiments, analyzed the data, prepared figures and/or tables, authored or reviewed drafts of the article, and approved the final draft.
- Alexander Ziegler conceived and designed the experiments, performed the experiments, prepared figures and/or tables, authored or reviewed drafts of the article, and approved the final draft.

### Field Study Permissions

The following information was supplied relating to field study approvals (*i.e.*, approving body and any reference numbers):

Field collection was approved by the National System of Conservation Areas (Costa Rica; SINAC-SE-CUSBSE-PI-R-131-2016) as well as the Comision Nacional de Acuacultura y Pesca (Mexico; PPF/DGOPA-291/17).

### Data Availability

The data is available at MorphoBank: http://dx.doi.org/10.7934/P4915.

## Supplemental Information

Supplemental information for this article can be found online at http://dx.doi.org/10.7717/peerj.17178#supplemental-information.

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
