# Peer review of "Selective concentration of iron, titanium, and zirconium substrate minerals within Gregory’s diverticulum, an organ unique to derived sand dollars (Echinoidea: Scutelliformes)"

_PeerJ, doi:10.7717/peerj.17178_

## Round 0.1 · original submission · Major Revisions

Dear authors, your manuscript reviewing and expounding upon a interesting phenomenon of echinoid biology. It is of interest to a broad range of scientist including paleontologist, biologist, specialists in biomineralization and sedimentology.

As noted in one of reviewer, your MS needs expansion of biological/ecological implications in the introduction and discussion. The research question needs to be more succinct and needs a greater emphasis on the ecology and evolutionary history of this group of sand dollars to be of interest to fill in the knowledge gap.
Although two reviewers proposed a minor revision to the manuscript, a third reviewer noted very serious deficiencies that should be corrected. Therefore, I believe that the manuscript needs a major revision. Therefore, I ask you to pay special attention to his comments.

When preparing your revised manuscript, you are asked to carefully consider the reviewers comments which are attached, and submit a list of responses to the comments. Your list of responses should be uploaded as a file in addition to your revised manuscript.

Reviewer 1 ·

Basic reporting

This is well prepared and clear represented experimental manuscript that can be accepted for publication after minor revision. The data represented of this work is of fundamental importance because our knowledge about biological reason of minerals deposition in the form of selected grains is still unknown. The analytical methods used in the study are of acceptance. However, more detailed EDX analysis of zircon grains will show also the occurrence of up to 4% Hf.

Experimental design

The analytical methods used in the study are of acceptance.
Research questions are well defined. Ethical standards are of acceptance.

Validity of the findings

This is well prepared and clear represented experimental manuscript that can be accepted for publication after minor revision. The data represented of this work is of fundamental importance because our knowledge about biological reason of minerals deposition in the form of selected grains is still unknown.

Additional comments

The authors cited three papers by Elkin & Co. Unfortunately, we cannot accept their results because these publications have not been peer-reviewed. We know this group personally and have studied the same grains. We plan to publish corresponding paper in the near future with results, which will differ from those already published in Russia.
Critical remarks:
- The is lack on clear formulated outlook

Reviewer 2 ·

Basic reporting

This is a nice paper reviewing and expounding upon a classic phenomenon of echinoid biology. It is of interest to a broad range of natural scientist including biologist, paleontologist, mineralogists and sedimentologist.

Ingestion of heavy minerals is a well-known, although somewhat peculiar features of sand dollars. It has never been studied at such details as in the submitted paper, which is thus of great interest to the echinoderm community as well as beyond.

Very well written using excellent literature resources. Broad methodology resulting in concise results with respect to mineralogy, grain sized distribution and selective feeding. A wide range of taxa are included. The paper is accompanied by excellent figures.

The classic paper and interpretation of Chia 1973 with respect to heavy minerals as a weight belt should be put into the introduction. It only shows up at the end of the paper in the conclusions Line 449, especially since the results seem to justify the original interpretation as a weight belt.

A general question: Any evidence for magnetotaxis? Perhaps directing the feeding patterns of sand dollars with respect to coastline direction. Though unlikely, this will be a factor what many readers will expect to be addressed.

Experimental design

Very thorough with a variety of appleid methods.

Validity of the findings

The findings are well supported by the experimental set up.

Additional comments

Some additional points:
Line 75: Mexico mentioned as approved collection site, but there no mention of where the Mexican samples were collected (should be added to line 75).
Lines 96-97: Which set of sieve sizes were used?
Line 105: The presence is aragonite is unexpected. Which skeletal elements of these sand dollars have an aragonitic skeleton, or does this represent ingested shell material from mollusks and the like.
Line 108 and others: At which institutes were the apparatuses that were used for the analysis (Siemens Diffractometer, SEM, X ray imaging etc.)?
Lines 378, 438: It is not clear what “opportunistic” (Lines 378, 438) means. Opportunistic in what respect and opposed to what other kind of behavior?

Reviewer 3 ·

Basic reporting

This paper is well written in parts and uses clear and unambiguous English but needs focusing throughout. Good references are used that provide sufficient background although a few could be added (e.g. Harold & Telford (1990), Systematics, phylogeny and biogeography of the genus Mellita (Echinoidea: Clypeasteroida), Journal of Natural History , vol. 24).

The figures are of high quality, although the distribution map would be aided by using different colours for Mellitella and Encope which are both green and have overlapping distributions. It would also be clearer if Dendraster (Pacific) and Leodia (Atlantic) were differentiated by colour (currently both red), even though they are geographically separate.

The raw data has been shared and the article is of a professional structure. The results are relevant to the hypotheses but could be tightened-up.

Experimental design

This paper is interesting to a specific audience, but needs expansion of biological/ecological implications in the introduction and discussion. The paper needs focusing throughout. The research question needs to be more succinct and in my opinion needs a greater emphasis on the ecology and evolutionary history of this derived group of sand dollars to be of interest to fill in the knowledge gap. Why do certain species retain specific minerals in Gregory’s diverticulum, how does this relate to the energetic environment and the substrate composition? Does this limit the distribution of specific-species to sand that is terrigenous in origin rather than biogenic and therefore lighter? Why is there variation within species? Answers are often in the text and are in the conclusion but this could be made more obvious to the reader earlier.

Why do only "most" (line 42) extant scutelliform sand dollars possess Gregory’s diverticulum. What is the authors hypothesis as to why certain species don’t possess this organ? How is this reflected by divergence within the group, habitat selection and ecology? Chia (1973) suggested that juvenile Dendraster excentricus selectively ingest heavy sand grains from the substrate and store them in an intestinal diverticulum which may function as a weight belt, but that this is lost after they reach 30 mm. How do the authors reconcile this finding or don’t they? Mooi & Chen (1996) interpreted the presence of the weight belt character in the Clypeasteroida as a result of phylogenetic history, and not of adaptive factors. This is only first mentioned in the conclusion and should come earlier.

The focus on non-destructive sampling and measurement is good, particularly for museum specimens but some of the key findings are missing from the abstract

Significance is mentioned in the text without statistics to support this.

Line 72. Various sources is too vague, refer to table stating where each specimen was sourced and if in collections the accession numbers.

Line 89. If you are washing with hydrogen peroxide (commercial bleach) aren’t you potentially removing any calcium carbonate components of the ingested particles?

Line 91. How did you isolate the grains from the diverticulum?

Validity of the findings

The results need to be written in a way that they clearly answer the research questions. The information is there, it just needs more focused reporting. Some statistics are missing.

Results
Line 148. Needs to refer to a graph or data analysis. If this is significant state significance level. What statistics were carried out to assess this? I don’t see these in the methodology.
Line 164. Give specifics of diverticular grains for each species or at least refer to the table. As written this is too vague.

Discussion
Line 235. This need stating in the abstract as a major finding.
Line 251. Similar here.
Do we need a justification for the methodological approach in the discussion? Surely just an appraisal of the best methods depending on what the objective of a future study is.
Line 331 to 332. Again needs to go in abstract. “The particular mix of mineralogy for any given species appears to be more a function of the heavy mineral composition of the substrate than any particular mineral selectivity on the part of the animal”

Line 357. This suggests that selectivity of grain size does occur through development. How do you know this is not significant?

Line 365. If retention is a passive process, how important is active selection by tubefeet?
Line 367 should be pedicellariae (pedicellaria is singular). Why are pedicellariae involved in ingesting particles? They would just clean sand particles from the surface of the sand dollar.
Line 369. How does substrate and particle size link to habitat? For example Leodia is limited to biogenic sand of the Atlantic and Mellita to terrigenous sands of the Atlantic and eastern Pacific (Harold & Telford, 1990).

Conclusion
The conclusion is well written and sums up the findings of the research. However, this is the first time some ideas are discussed.

Some of these findings need to go in the abstract, i.e. "... selection of a particular mineral grain by a sand dollar is opportunistic, as long as the grain has a high specific gravity"

Additional comments

This paper contains a lot of interesting findings but needs rewriting so that it logically leads to the authors conclusions. The abstract is missing important findings, the introductions doesn't mention enough ecology and the idea that heavy minerals are retained as weight belts, and the research questions need to be more succinct. The paper could be shortened and written in a way that leads the reader to the authors conclusions.

---

## Round 0.2 · accepted · Accept

Dear Drs. Zachos and Ziegler,

I have assessed the revision myself, and I am happy with the current version. Thank you for the detailed responses to the comments of the reviewers and for the corrections made to the manuscript.

Now your manuscript is ready for publication.

Best regards,
Alexander